META-RESEARCH ARTICLE

# Systematically assessing microbiome–disease associations identifies drivers of inconsistency in metagenomic research

Braden T. Tierney[1,2,3,4], Yingxuan Tan[1¤], Zhen Yang[2,3,4], Bing Shui[5], Michaela J. Walker[6], Benjamin M. Kent[7], Aleksandar D. Kostic[2,3,4]*, Chirag J. Patel[1]*

**1** Department of Biomedical Informatics, Harvard Medical School, Boston, Massachusetts, United States of America, **2** Section on Pathophysiology and Molecular Pharmacology, Joslin Diabetes Center, Boston, Massachusetts, United States of America, **3** Section on Islet Cell and Regenerative Biology, Joslin Diabetes Center, Boston, Massachusetts, United States of America, **4** Department of Microbiology and Immunobiology, Harvard Medical School, Boston, Massachusetts, United States of America, **5** Department of Cancer Biology, Dana Farber Cancer Institute, Boston, Massachusetts, United States of America, **6** UPSIDE Foods, Berkeley, California, United States of America, **7** US Marine Corps, Camp Pendleton, California, United States of America,

¤ Current address: Department of Computer Science, Stanford University, Stanford California, United States of America

* Aleksandar.Kostic@joslin.harvard.edu (ADK); chirag_patel@hms.harvard.edu (CJP)

**Academic Editors:** Jonathan A. Eisen, University of California Davis, UNITED STATES and Marcus Munafò, University of Bristol, UNITED KINGDOM

**Data Availability Statement:** Scripts used in the analysis (i.e. to generate figures and download

## Abstract

Evaluating the relationship between the human gut microbiome and disease requires computing reliable statistical associations. Here, using millions of different association modeling strategies, we evaluated the consistency—or robustness—of microbiome-based disease indicators for 6 prevalent and well-studied phenotypes (across 15 public cohorts and 2,343 individuals). We were able to discriminate between analytically robust versus nonrobust results. In many cases, different models yielded contradictory associations for the same taxon–disease pairing, some showing positive correlations and others negative. When querying a subset of 581 microbe–disease associations that have been previously reported in the literature, 1 out of 3 taxa demonstrated substantial inconsistency in association sign. Notably, >90% of published findings for type 1 diabetes (T1D) and type 2 diabetes (T2D) were particularly nonrobust in this regard. We additionally quantified how potential confounders—sequencing depth, glucose levels, cholesterol, and body mass index, for example—influenced associations, analyzing how these variables affect the ostensible correlation between *Faecalibacterium prausnitzii* abundance and a healthy gut. Overall, we propose our approach as a method to maximize confidence when prioritizing findings that emerge from microbiome association studies.

## Introduction

With its role in host health, the microbiome field is rapidly accelerating toward the clinic in the form of new diagnostics and therapeutics. An instrumental first step toward this lofty goal, however, is vetting individual microbial features (e.g., species abundance) for their association with disease. These microbiome association studies (MAS, i.e., identifying sets of microbiome

literature review information) can be found at
https://github.com/chiragjp/ubiome_robustness.
All the relevant microbiome datasets can be
downloaded from the R package associated with
curatedMetagenomicData [Pasolli et al. Accessible,
curated metagenomic data through
ExperimentHub. Nat Methods. 2017;14: 1023–
1024. ]. All analyzed data from this study can be
located at https://figshare.com/projects/
Microbiome_robustness/127607.

**Funding:** CJP and BTT were funded by the National
Science Foundation (#1636870), the National
Institute of Allergy and Infectious Diseases
(R01AI127250), and the National Institute of Health
Sciences (R01ES032470). ADK was funded by an
American Diabetes Association Pathway to Stop
Diabetes Initiator Award (#1-17-INI-13) and the
Smith Family Foundation Award for Excellence in
Biomedical Research. The funders had no role in
study design, data collection and analysis, decision
to publish, or preparation of the manuscript.

**Competing interests:** We have read the journal's
policy and the authors of the manuscript have the
following competing interests: ADK is a co-founder
of FitBiomics, Inc. and a member of their Scientific
Advisory Board. BTT consults for Seed Health on
microbiome study design and analysis. MJW is an
employee of UPSIDE Foods. UPSIDE Foods, Seed
Health, and FitBiomics were not involved in the
funding or preparation of this manuscript or the
work described therein.

**Abbreviations:** ACVD, atherosclerotic
cardiovascular disease; BY, Benjamini–Yekutieli;
CIRR, cirrhosis; CLR, center logged ratio; CRC,
colorectal cancer; FDR, false discovery rate;
HSCRP, high-sensitivity C-reactive protein; IBD,
inflammatory bowel disease; MAS, microbiome
association studies; T1D, type 1 diabetes; T2D,
type 2 diabetes; VoE, vibration of effects.

features in observational datasets that are correlated with disease presence) have been critical
in generating beaucoup hypotheses regarding the role of, specifically, the gut microbiome in
many diseases, from autoimmune disorders to cancers [1,2]. However, MAS are subject to a
litany of biases. From DNA extraction method to bioinformatics pipelines, every choice a
researcher makes can potentially influence the conclusions derived from an observational
analysis.

Additionally, variation in modeling strategy can lead to analogously varying findings [3,4].
One specific form of bias that can affect results is confounding, or not accounting or stratifying
for variables that are associated with both a given microbial feature and a disease phenotype of
interest [5]. Accounting for confounding in the microbiome space is particularly difficult due
to the sheer volume of variables that can potentially affect microbiome composition as well as
the millions of possible features (e.g., taxa, pathways, and genes) that can be identified in
microbiomes.[6,7].

An analytic approach to address confounding includes adjusting by an a priori selection of
potential confounders. Many choose a bespoke set of variables to control for a priori hypothe-
sized confounding. However, this is a choice that must be justified. Sensitivity analyses (which
fall under the broader umbrella of "multiverse" analyses) [8,9] allow investigators to explore
the space of analytic choices (i.e., what specific variables to adjust for) that may influence
modeling outcomes. These outcomes may include association sizes, predictions, and *p*-values,
which can vary depending on modeling strategy, sampling size, and measurement error [3,10–
12]. These analyses may be particularly useful for discovery-based studies (very common in
the microbiome and genomic fields), approaches designed to generate, rather than test specific
candidate, hypotheses from complex datasets. We refer to the distribution of possible associa-
tions that emerge from different modeling scenarios when carrying out discovery as vibration
of effects (VoE) [3,13–15]. In some cases, slight changes to model specification yield polar
opposite results (e.g., a particular microbiome feature being both negatively and positively
associated with disease) [16]. In this manuscript, VoE is computed by, for each microbial fea-
ture–disease pairing, fitting all possible linear models, each adjusted by different features,
while tabulating how the association between the microbial feature and disease changes.
Robust microbial feature–disease associations are those whose association size does not change
too much with respect to the number and type of adjustment variables in the model.

To date, the immense impact model choice and confounding can have on the microbiome
has only been investigated in some isolated cases. For example, Forslund and colleagues found
that patient use of metformin—a common antidiabetic medication—confounded the associa-
tion between type 2 diabetes (T2D) and gut microbiome features, generating misleading and
difficult-to-interpret conclusions [2]. Vieira-Silva and colleagues demonstrated that statins, a
common cholesterol lowering therapeutic, confound associations between BMI and the gut
microbiota [17]. Similarly, other studies have identified that certain features, like age and stool
consistency, can confound associations with host phenotypes and microbiome data [18–21].
Most recently, Vujkovic-Cvijin demonstrated that dietary variables, age, sex, and BMI can
confound associations between a range of diseases and the gut microbiome [6]. These studies,
however, consider limited and candidate groups of potential confounding adjustment vari-
ables, and they do not systematically assess how sets of confounders or varying study designs
(when considered together or separately) influence association size and direction (e.g., associ-
ated with risk for disease or protective of disease) across published results.

Here, to gauge the impact of model specification in MAS, we deploy a systematic sensitivity
analysis, measuring VoE in reported microbiome associations. Comparing modeling strate-
gies, we quantify the robustness (variation as a function of model specification) in microbial
taxon–disease associations across 6 different phenotypes. We counted how many associations

(published and otherwise) are recovered (e.g., appear as statistically significant) or lost when undergoing sensitivity analysis. We propose modeling VoE as one of many potential steps in building association prioritization frameworks, metrics for prioritizing microbiome findings for in vivo validation.

## Results

### Meta-analysis combined with modeling vibration of effects recovers and prioritizes associations

From an existing database [22] of public metagenomic shotgun sequencing data, we accessed 15 studies comprising samples from patients with 6 diseases: colorectal cancer (CRC), type 1 diabetes (T1D), T2D, atherosclerotic cardiovascular disease (ACVD), inflammatory bowel disease (IBD), and liver cirrhosis (CIRR) (**Fig 1A and 1B**). We built a database of taxa that were prevalent (in >10% of samples) in each study. We then searched the literature, taxon by taxon, for reports of each feature being associated with any of the 6 diseases (see Methods). Importantly, many (214, 37.8%) of these findings were directly from papers present in the data used in this study (in the case of CRC, ACVD, CIRR, T2D, and T1D; **S1 Table**).

We computed initial, univariate associations for each taxon in our aggregated dataset. We additionally benchmarked the data transformation and modeling strategies underlying these associations (**S1 Text, S1 Fig**). We refer to this model, which contains the phenotypic variable of interest as the sole covariate, as the baseline model (**S1 Table**). Three of these diseases (T2D, T1D, and CRC) had data spread across multiple cohorts. For these, we meta-analyzed across individual associations within each cohort to compute overall summary statistics. We found a

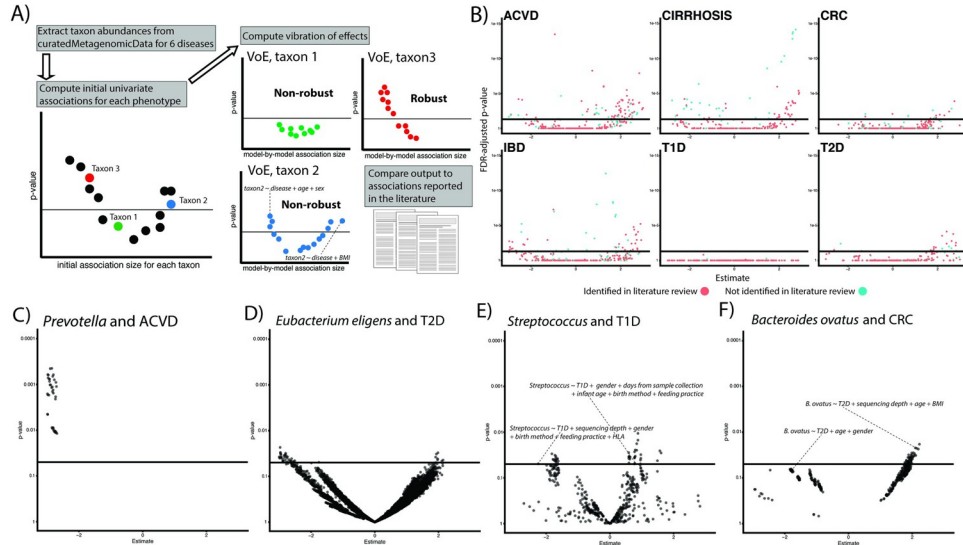

**Fig 1. (A)** Overview of approach. We extract prevalent microbial features from our datasets and attempt to reproduce the findings from the literature by modeling VoE. We additionally review the literature for reported gut microbiome associations (their reported direction of correlation) with 6 diseases of interest. **(B)** Volcano plots showing the output from the initial, univariate associations. Point color corresponds to if an association was identified in our literature review solid line represents FDR significance (adjusted $p < 0.05$). **(C–F)** Examples of robust (C) and nonrobust associations. Each point represents a different modeling strategy. Solid line is nominal ($p < 0.05$) significance. This figure can be generated using the code deposited in https://github.com/chiragjp/ubiome_robustness and the data deposited in https://figshare.com/projects/Microbiome_robustness/127607. ACVD, atherosclerotic cardiovascular disease; CIRR, cirrhosis; CRC, colorectal cancer; FDR, false discovery rate; IBD, inflammatory bowel disease; T1D, type 1 diabetes; T2D, type 2 diabetes; VoE, vibration of effects.

total of 720 features that were statistically significant after adjusting for false discovery rate (FDR), 199 (24.8%) of which were reported in the literature (**Fig 1B**). The number of significant features was dependent on phenotype and number of cohort analyzed; for example, T1D had no statistically significant results, CRC had 29 (52.7% of which were reported in the literature), and CIRR had 298 (25% of which were reported in the literature).

We next executed a systematic VoE analysis, fitting a total of 6,035,110 models, each employing multiple linear regression with microbial feature abundances as the dependent variable (**S1 Table** contains information on the number and type of covariates per disease). Some associations (e.g., **Fig 1C**) were robust; that is to say, regardless of modeling strategy they yielded similar results (e.g., same direction of association across models). Others, however, demonstrated striking variation in output as a function of modeling strategy (e.g., **Fig 1D**). In some cases, very similar models (e.g., only different by inclusion/exclusion of a single variable or 2 in the model) yielded opposite and nominally significant results (**Fig 1E and 1F**).

We next aimed to stratify associations by their (1) presence in the literature; and (2) robustness/recovery as a function of vibration effects. A total of 509 features were FDR significant at baseline, additionally during VoE, and not reported in the literature review (**Fig 2A**, blue column). A total of 187 were FDR significant at baseline and reported in the literature (**Fig 2A**, orange column). An additional 61 of others (10% of all literature-based associations) found in the literature were FDR significant in at least 1 vibration but not in the baseline model (**Fig 2A**, red columns). This brought the total number of taxa–disease associations we were able to recover to 248. Additionally, 264 taxa were not found in the literature but were FDR significant at least once during a vibration (**Fig 2A**, green columns). In other words, we observed that modeling VoE was able to shed light on associations that would be potentially overlooked by single modeling strategies, in some cases recovering results reported in the literature.

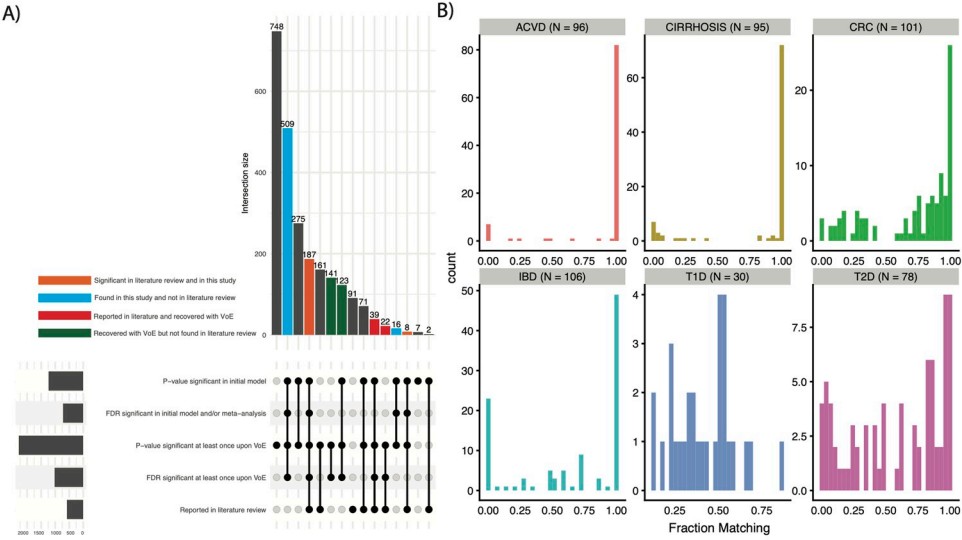

**Fig 2. Comparing single modeling approaches to modeling VoE. (A)** Stratifying features by their being identified in our literature review, *p*-value/FDR significant (<0.05) in our initial analysis, or having at least 1 significant model upon vibrations. The gray bars labeled "set size" indicate the number of features associated with a given row in the bottom of the plot (e.g., about 1000 features were *p*-value significant in the initial model). The gray bars in the upper portion of the panel are those we chose not to highlight, as they do not fall into any category indicated by the colors or referenced in the manuscript. **(B)** Distribution of fraction of models matching literature review–defined direction of associations. This figure can be generated using the code deposited in https://github.com/chiragjp/ubiome_robustness and the data deposited in https://figshare.com/projects/Microbiome_robustness/127607. ACVD, atherosclerotic cardiovascular disease; CRC, colorectal cancer; FDR, false discovery rate; IBD, inflammatory bowel disease; T1D, type 1 diabetes; T2D, type 2 diabetes; VoE, vibration of effects.

As another measure of robustness, for each taxon for each disease, we report the fraction of associations with signs matching the literature (Fig 2B). For all diseases except T1D, these distributions matched what was reported before was bimodal. The mode of the distribution was closer to 1, indicating a large frequency of high concordance associations and a moderate frequency of extremely low concordance (i.e., almost all models pointing the opposite direction as the literature) associations. Given the distribution of the data in Fig 2B, we defined a low concordance association as agreeing in direction in 50% of models fit. In total, 27.9% of all features fit into this category; in other words, 1 in 3 features were discordant with the literature in 50% of all vibrations.

## Association robustness varies as a function of disease and cohort

For the remainder of the analysis, we opted to prioritize (and thereby analyze) only the 581 findings that we identified as reported in the literature (Fig 2A, orange columns, S2 Table). The robustness (i.e., number of associations with consistent direction and significance) of prior-reported associations varied for different disease phenotypes (Fig 3). CIRR had the greatest fraction of associations that achieved initial FDR significance (64/106, 60.4%), followed by ACVD (47/96, 49.0%), followed by IBD (46/140, 32.9%). T1D and T2D contained the least robust associations, with 0 features (out of 34) and 5 features (out of 96, 5.2%) achieving FDR significance, respectively. For T2D, this is particularly surprising as the species we selected to analyze had been reported as significantly associated with disease even when adjusting for metformin usage.

Of the 325 features that, after vibrating, had at least 1 FDR significant model, 114 (35%) were highly nonrobust, with at least 20% of models conflicting in association direction. This was most striking for T1D and T2D (Fig 3, bottom row), where nearly all tested associations were nonrobust. The diseases for which we had only a single cohort dataset (ACVD, IBD, and CIRR; see Methods) had greater association consistency and higher fractions of statistically significant findings when compared to the multicohort, meta-analyzed, phenotypes. This indicates that the single cohort associations should be further validated/tested in other populations to confirm their robustness.

We next sought to probe researcher degrees of freedom: the probability that a statistically significant association would arise in the event that a researcher were to fit a single model instead of multiple. We calculated the fraction of FDR adjusted (using the cutoff from our prior analyses) and nominal $p$-values that were less than 0.05 for all models for a given feature from our literature review (Fig 3, S2 Fig). For all models for a given taxonomic feature, on average, 38% had nominally significant (or 16% FDR significant) $p$-values. Of the reported associations we tested, 488 (84.0%) had at least 1 model that was nominally significant, and 248 (42.6%) had at least 1 model that was FDR significant.

We additionally identified particular cases where our baseline modeling approach was clearly too conservative, initially excluding associations reported in the literature that, after modeling VoE, clearly should be of interest. For example, in the association between *Roseburia* and ACVD, the association sizes all pointed in the same direction, and 74/127 (58.6%) of models were FDR significant, despite the univariate association being not (S3 Fig). We examined the variables present in each model and identified that 64/74 of the FDR significant vibrations were adjusted for gender, whereas none of the nonsignificant models were.

## A vast array of possible adjusting covariates influence model output in microbiome associations

Motivated in part by this *Roseburia*—gender result and similar observations, we next aimed to identify the sources of VoE in our associations en masse, computing how variation in estimate

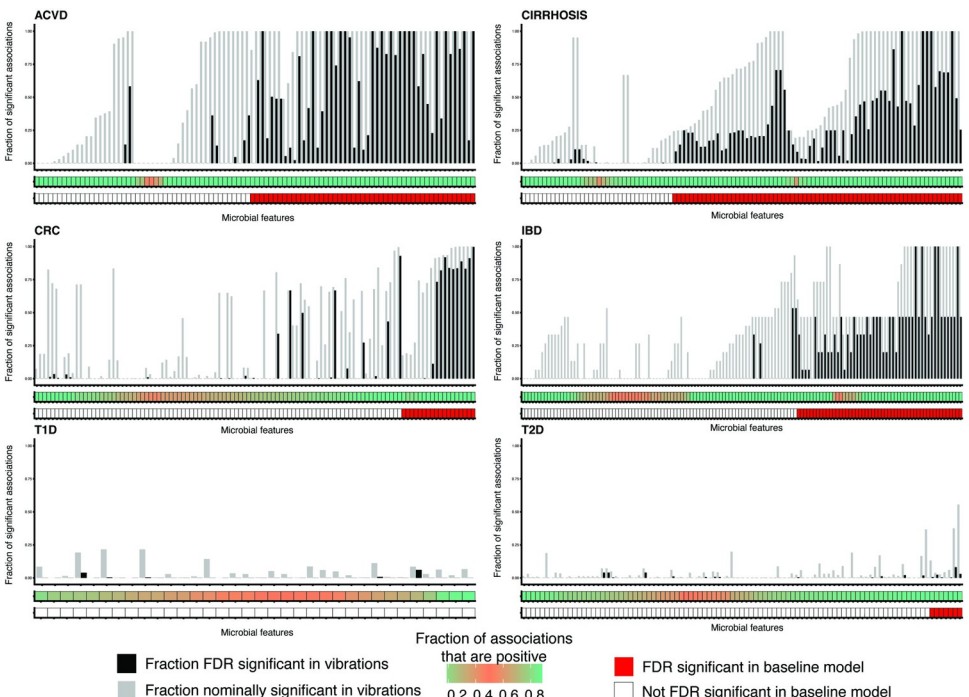

**Fig 3. VoE for reported associations from the literature in the form of summarized modeling output. (A)** We measure association robustness in part by computing the fraction of associations with signs greater than 0. These values show the distributions of these values for each microbial taxon). **(B)** Summarized modeling output. Red blocks indicate organisms that were FDR significant in our study. The middle bar describes the fraction of association sizes greater than 0 per taxon: a highly confounded association will be closer to 0.5 and pink, whereas more robust associations will be closer to 0 or 1 and green. The gray bars in the upper bar plot corresponds the fraction of models that were nominally (*p*-value < 0.05) significant for the microbial feature–disease association, whereas the black bars correspond to the fraction of models that were FDR significant. Features marked as significant in our study but never FDR significant were only significant after the meta-analysis and did not have any nominal significant *p*-values. See S2 Fig for this plot reproduced with species names on the x-axis. This figure can be generated using the code deposited in https://github.com/chiragjp/ubiome_robustness and the data deposited in https://figshare.com/projects/Microbiome_robustness/127607. ACVD, atherosclerotic cardiovascular disease; CRC, colorectal cancer; FDR, false discovery rate; IBD, inflammatory bowel disease; T1D, type 1 diabetes; T2D, type 2 diabetes; VoE, vibration of effects.

size can be attributed to the presence or absence of specific adjusting variables from a model. We used a mixed effect linear modeling approach (see Methods) to determine how associations with a given disease changed as a function of presence or absence of other adjusting variables (e.g., age, sex, and BMI). We hypothesized that this approach would identify different kinds of biasing adjusters, like confounder or collider variables. As a form of benchmarking, we estimated in the T2D cohort how the beta coefficients on the adjusting variables from the mixed modeling approach changed as a function of the number of vibrations executed (**S4 Fig**). We found 100,000 vibrations (our upper limit) to be sufficient to identify consistent correlation between these beta coefficients (Pearson >0.9).

Accounting for BMI, age, sequencing depth, and gender all had strong influence on both the size and direction of microbial feature–disease associations. While these variables altered the absolute value of association size across all diseases, the direction and order of magnitude of change (i.e., inflating or deflating associations) were, as expected, disease specific. Adjusting for gender, for example, consistently increased association sizes across diseases, with the only exception being CIRR. In the case of T2D, country of origin inflated association sizes, whereas it deflated association sizes for IBD and CRC.

T2D and T1D were influenced by a wide array of adjusting variables, some on the pathway for disease, others not. First, adjustment for glucose levels inflated association sizes for both diseases. We found feeding practice, as well as HLA genotype (**Fig 1E**), to have a substantial influence on associations. Feeding practice has been reported as a confounder of T1D [23]. T2D associations were also substantially affected by blood pressure, BMI, cholesterol, creatinine, and HbA1c. In prior studies [2,24], metformin usage also confounded associations; however, its effect was not as strong as some of the other adjusting variables (**Fig 4A**), such as adiponectin. Similarly, alcohol use appeared to bias CIRR associations and cholesterol levels influenced associations in ACVD. In **Fig 4B and 4D**, we indicate the impact of these and other adjustment variables visually, showing which models accounted for them during the vibration analysis. For example, models including delivery mode or feeding practice in *Streptococcus* and T1D associations tended to yield positive associations (e.g., an increase in *Streptococcus* associated with higher T1D odds), whereas adjusting for cholesterol yielded negative associations (an increase in *Streptococcus* associated with lower T1D odds). Similarly, in T2D, adjusting for HbA1c in *Roseburia* associations yielded positive associations, whereas adjusting for creatinine yielded the opposite, on average.

## Vibration of effects reveals disease-specific variation in *Faecalibacterium prausnitzii* associations

We additionally took interest in the microbe *F. prausnitzii*, as it was reported in the literature [25–29] as negatively associated with 5 out of 6 diseases except T1D. We found this negative association to be highly robust for 3/5 diseases; however, CRC and T2D exhibited notable inconsistency in association direction (**Fig 5A**).

We identified that the association between T2D and *F. prausnitzii* tended to be positive when we adjusted for glucose, high-sensitivity C-reactive protein (HSCRP), and cholesterol (**Fig 5B**). For CRC, the association tended to appear positive when models were adjusted for cholesterol (**Fig 5C**). It should be noted that it is possible that some of these variables could be colliders (e.g., BMI and T2D). Comparably, the immediately robust associations in the other 3 diseases should be treated with higher priority and perhaps are worthy of validation in wet lab experiments.

## Discussion

In this study, we explored the utility of modeling VoE for MAS. Across a number of diseases with varying number of cohorts, sample sizes, and variables, we showed how massive-scale, automated sensitivity analysis can be used to (1) query how associations found in the literature may change as a function of model choice; (2) identify overlooked associations (such as the association between *Roseburia* and ACVD, which a single modeling approach may miss); and (3) identify sets of potentially important, disease-specific confounding variables that should be accounted for in current and future MAS. We do not claim that our approach is the be-all-and-end-all of microbiome statistical sensitivity analysis. It is merely one approach for quantifying association robustness. It is for this reason that we hesitate to claim that we are holistically measuring reproducibility, as the requirements for rigorously doing so are nebulous [30] and would likely require considering all analytic biases that pervade microbiome analyses, something outside the scope of this project. For example, to assess reproducibility in totality, we should have recreated modeling strategies (e.g., using particular adjusting variables or different regression approaches) used in published results as opposed to instead fitting univariate models.

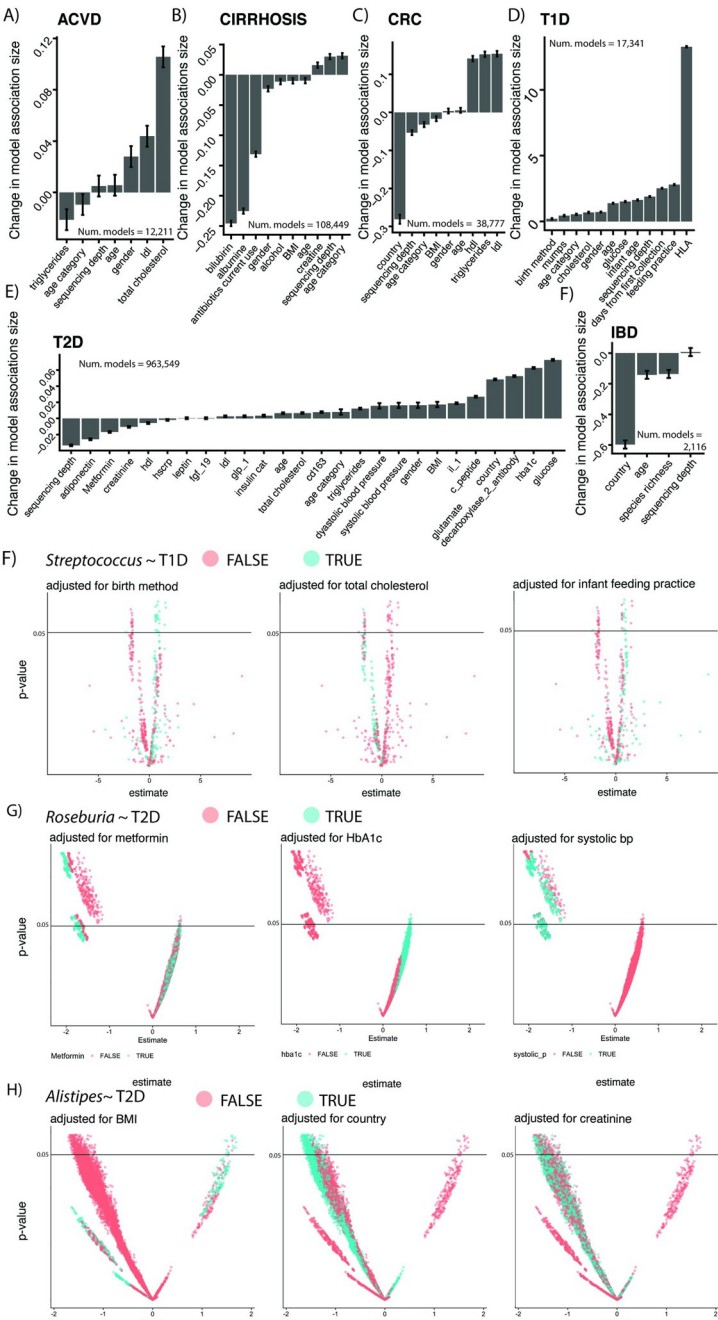

**Fig 4. The effects of different adjusters on human microbiome associations. (A)** Various adjusters for our diseases of interest. For each disease in our study, we report the change in the association sizes between microbiome features and disease as a function of adjusting variable presence or absence (See Methods). Each individual plot summarizes the output for the 2^n models fit for each feature within a given disease, where *n* = number of adjusters. The y-axis corresponds to the mean change in Beta coefficient (in units of relative abundance) on the independent, binary disease outcome when a given adjusting variable (x-axis) is included in the model. **(B–D)** Visualization of the impact of the presence/absence of different confounders for 3 organisms and their associations with T1D/T2D. This figure can be generated using the code deposited in https://github.com/chiragjp/ubiome_robustness and the data deposited in https://figshare.com/projects/Microbiome_robustness/127607. ACVD, atherosclerotic cardiovascular disease; BP, blood pressure; CRC, colorectal cancer; IBD, inflammatory bowel disease; T1D, type 1 diabetes; T2D, type 2 diabetes.

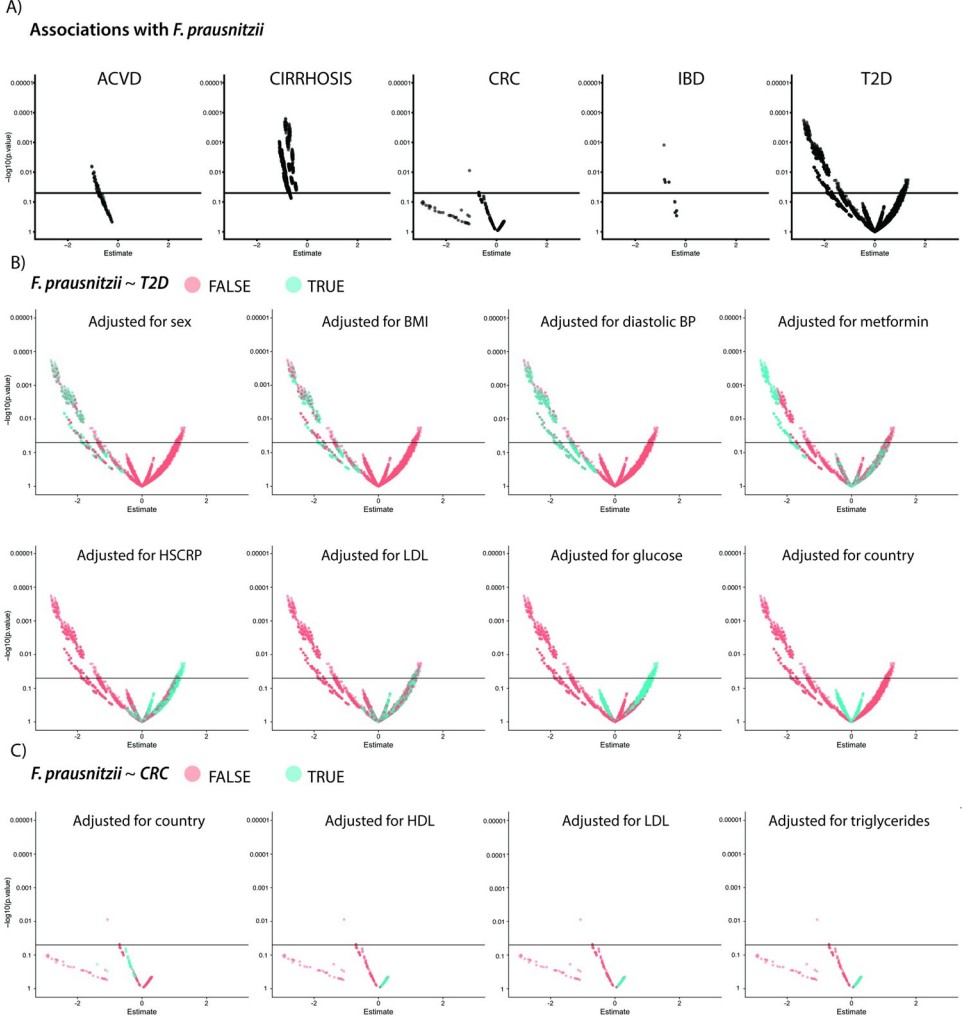

**Fig 5. Exploring the impact of variable adjustment strategies on *F. prausnitzii* ~ disease associations. (A)** VoE for the 5 phenotypes that had associations with *F. prausnitzii* reported in the literature. **(B, C)** The impact of variable adjusting strategies for (B) T2D and (C) CRC. This figure can be generated using the code deposited in https://github.com/chiragjp/ubiome_robustness and the data deposited in https://figshare.com/projects/Microbiome_robustness/127607. ACVD, atherosclerotic cardiovascular disease; BP, blood pressure; CRC, colorectal cancer; HDL, high-density lipoprotein; HSCRP, high-sensitivity C-reactive protein; IBD, inflammatory bowel disease; LDL, low-density lipoprotein; T2D, type 2 diabetes; VoE, vibration of effects.

A few microbiome–disease associations, such as *Helicobacter pylori* and gastric cancer [31], *Akkermansia muciniphila* and obesity [32], and *Fusobacterium nucleatum* and CRC [33], have been replicated in multiple studies or experimentally validated (e.g., in animal models or clinical trials). Until we have a framework for the sensitivity of the resultant observational host disease–microbiome associations, the thousands of associations published will be of limited value in experimental settings.

We claim that one step—out of many—toward translating microbiome findings into biological understanding is determining how best to prioritize for future (e.g., in vivo) investigation associations arising from MAS. There is a need for association prioritization frameworks: the contexts in which of these manifold associations are most worthy of wet lab experimentation [34–36]. In this study, we do so by identifying which associations are consistent in direction and statistically significant across multiple cohorts. Furthermore, we propose modeling

VoE to pressure test existing associations, and, sidestep reliance on single models or metrics (e.g., *p*-values), potentially recovering or identifying features that would be overlooked with only one modeling approach (as in **S3 Fig**). Further, VoE provides a path forward for identifying adjusting variables that may influence association size and direction.

That said, modeling VoE is certainly not the only way to identify an association worth prioritizing. Furthermore, an ostensibly robust association viewed only through the lens of VoE still could be a false positive or dependent on, for example, data processing pipeline choice (e.g., the decision to average repeated measures data versus selecting one sample per individual) [37]. A more comprehensive framework could rely on a number of heuristics, for example, putting the greatest emphasis on associations that have the best model fit, are reported across multiple large cohorts, and/or have undergone sensitivity analysis via VoE.

Of course, our approach is not without other drawbacks. First, we did not preregister our study, which would set an even higher bar for reproducible results. This is particularly difficult for hypothesis-generating studies, though, as the outcomes cannot be preset—for example, our decision to focus on *F. prausnitzii* as a case study could not have been made until seeing the results.

Second, as we observed in **S1 Fig**, vibration may be contingent on the number and type of variables measured, the size of the cohort, and measurement error. Further, sources of variables that influence associations that we identified could themselves hold interesting biological signals. For example, diet, BMI, and cholesterol had strong influence on associations and are worthy of further investigation. Specifically, these variables may interact with microbes in their associations with disease (e.g., m higher BMI and presence of a microbe may have larger associations than the association across all BMI groups).

Third, our approach, at present, relies on exclusively linear modeling due to its speed and the ease of performing statistical inference on its association. Many microbiome studies have adopted random forests, which can capture nonlinearities but can be difficult to perform inference on or interpret individual variables [38–40]. Moreover, we ourselves made choices in our modeling strategy that could influence statistical power such as averaging repeated samples from the same individual. Finally, associations can also vary due to data processing choices, like different approaches quantifying taxon abundances, such log transforming or discretizing, yielding more or less reproducible results. Study design characteristics, such as error in the measurement, inclusion criteria of individuals, and sample size, will also influence associations. Alternatively, different studies use different methods or cutoffs for computing FDR (e.g., 0.15 or 0.07) [41,42]. Many of the associations that we extracted (214 of the 581) came directly from the cohorts analyzed here; therefore, disparities in associations, we claim, can be attributed to modeling approach and parameters and not to other differences in study design, such as sample size.

Finally, our approach does not in any way address causality. Indeed, many of the adjusters we identify could be on the pathway for a human disease, or a mediator [43], as in the case of alcohol usage and liver CIRR or BMI and T2D. In this sense, it is crucial to note that modeling VoE with this approach does not distinguish between certain forms of bias—like collider bias —that could be mistaken for confounding. As we stated before, however, this application of modeling VoE is meant to only identify potential variables of interest that are worth consideration and may or may not be confounders.

Notably, the systematic nature of VoE stands apart from many traditional modeling approaches. Instead of finding one correct model with a priori knowledge, it assumes a limited theoretical basis for how to model a question (e.g., what adjusters to include). While it is useful young fields like the microbiome, it may also introduce other biases (e.g., colliders) as a function of its use and may not be relevant for all disciplines, especially those steeped in theory

(e.g., economics). It complements existing approaches, such as Bayesian model averaging [44], whose primary goal is to provide an optimal single predictor by averaging across the many different models. Therefore, we posit here that VoE, which in future work should be compared to these other methods, should at present be used primarily as a way to probe associations from different modeling strategies to systematically assess the combination of potential adjustments. Further still, models need to be checked to ensure that they do not violate any assumptions. For example, we claim that our *F. prausnitzii* results indicate the need for further exploration to determine the biological relationship between *F. prausnitzii* and glucose levels.

Fundamentally, it is a broad scientific challenge to prioritize the best findings from observational studies in human populations to test their causal mechanism in an experimental setting. The history of science in this field is long, dating back from Fisher [45] to Bradford Hill [46] and to the current day of counterfactual and causal modeling.[47] However, these heuristics or approaches have yet to be extended to high-throughput observational research (many input variables and outcome variables) such as microbiome research [48]. Further still, criteria such as the Bradford Hill criteria, may not be compatible with high-throughput research. For example, Bradford Hill criteria such as "strength of association" may be impossible to fulfill, as microbiome–human disease association sizes are generally small and sensitive to adjustments.

In short, fitting and reporting a single model that encapsulates a few assumptions will not be conducive to efficiently and consistently delivering clinically actionable biology from microbiome association studies. However, if—in part by modeling VoE—we are able to identify robust-to-model assumption associations that reproduce across cohorts, we are one step closer to achieving clinical relevance for microbiome-based diagnostics and a deeper understanding of the role of the microbiome in human biology.

## Methods

### Data collection

curated Metagenomic DataV1.14.1 [22] contains a large number of shotgun sequencing microbiome datasets preprocessed with HUMAnN2 [49] output for each sample. We used this dataset for our analysis, using only the taxa-level (MetaPhlAn2) data. We selected for diseases that had either greater than 100 samples or had multiple cohorts, excluding 2—infectious gastroenteritis and otitis—due to their limited presence in the microbiome literature. This resulted in our including 2,343 samples, 15 cohorts, and 6 diseases (CRC, T1D, T2D, ACVD, IBD, and liver CIRR).

We would additionally like to note that our study was not preregistered. While this is typical for both microbiome studies and hypothesis generating (versus hypothesis testing) studies in general, we do acknowledge (and we thank a reviewer of this manuscript for pointing this out) that it would be useful for these metascience applications going forward. Indeed, given that our literature review involved searching for single reports of microbial species already in our dataset associated with disease (as opposed to looking for all possible reported associations), some form of preregistration could have clarified this approach. Going forward, preregistration would set an even higher bar for robust MAS.

### Literature review

We used PubMed as the source of our literature review. We did not aim to make this literature systematic in the sense that we were searching for every reported finding in the literature across multiple databases. Rather, in order to target our core question (what fraction of reported associations are reproducible in the datasets we had gathered), we were interested strictly if a given species found in curated metagenomic data (with greater than 0% abundance

in at least 10% of samples for a given cohort) had been reported in the literature as ever associated with 1 of our 6 diseases of interest. As a result, we took the following general approach:

1. We used NCBI's Entrez search utility to download information on all papers matching disease-specific criteria. This script is available at https://github.com/chiragjp/ubiome_robustness.

2. We then filtered out papers that were either not in human systems or involved clinical endpoints not relevant to the disease of interest (e.g., the association between the microbiome and a specific disease treatment).

3. We then read the remaining publications and looked specifically for reports that the microbes present in curatedMetagenomicData were associated with a given disease. Upon identifying a given association, we recorded the microbe, the direction of association, and the source paper. We opted to not record all of the different sources in which a microbe had been reported in the literature, instead only recording the reference where we initially encountered it. The exception to this rule was if we found conflicting reports of the direction of the association (i.e., positively correlated with disease, negatively in another). In this case, we recorded the second publication that reported the opposite sign association.

   In the case of reviews, we reported the associations the review described, recording as well the publications cited by the review that contained the original reporting of a given association.

   We had additionally specific subprotocols for the phenotypes in certain cases, which we detail in the following 4 blocks of text:

T2D: Given the reported confounding between T2D and metformin and massive variation in modeling strategies reported in the literature, we specifically recorded not just the initial association we encountered, but all of them found within our downloaded papers from PubMed. We then only included associations that had been reported consistently as either (1) adjusted for metformin usage across all studies reporting them or (2) consistently not adjusted for metformin across all studies. We did not include microbes that were, for example, associated with T2D in one study, but not found to be associated with T2D in another study where the modeling strategy was adjusted for metformin.

IBD: We included in our analysis microbes associated with either Crohn disease, Ulcerative Colitis, or both.

CIRR: We only included cohorts that included patients with the same description as those in the paper our data came from [50]. This was limited to patients with any kind of liver CIRR, but it excluded those where the disease had advanced to or was comorbid with other liver conditions (e.g., carcinoma).

T1D: Our metadata indicated if patients were ever diabetic or only healthy. Given that we chose to average across samples, creating one sample for each individual, we opted to only select findings from the literature that compared T1D cases to T1D controls. We did not, for example, include studies that attempted to compare T1D cases prior to onset to healthy individuals.

## Modeling vibration of effects

We used the quantvoe package (https://github.com/chiragjp/quantvoe) to compute associations, meta-analyses, and model VoE for each disease–microbe pair of interest. We first

computed an initial, univariate association for each pair. These took the form of a standard linear regression, ln (*microbial_feature* + *f*) ~ *disease*, where the *disease* variable is a binary variable indicating disease status and the *microbial_feature* variable corresponds to the relative abundance of a particular taxon. *f* is a fudge factor of 0.00000001 to account for 0 values prior to logging our data. For diseases with repeated sampling per individuals (T1D and IBD), we computed the average abundance of each feature within an individual during the entire observation window. For diseases found in multiple cohorts (T1D, T2D, and CRC), we computed a random effects meta-analysis using R's metafor [51] package over the initial association outputs (estimates and standard errors) for each input cohort (parameters: comb. fixed = FALSE, comb. random = TRUE, method. tau = 'REML',hakn = FALSE, prediction = TRUE, sm = "SMD", control = list(maxiter = 1000)).

After computing these initial associations, we adjusted for FDR across all 6 diseases simultaneously (i.e., generating just 1 cutoff instead of 6) using the Benjamini–Yekutieli (BY) method, selecting 0.05 as our significance threshold.

We then computed the VoE for each microbial feature—disease pairing of interest by fitting an array of models with varying specification strategies (e.g., differing covariates). We used clinical/human phenotypic covariates present in curated metagenomic data's combined metadata file as our database of adjusting variables. We only used those for which we had recorded information for 90% of samples. We only allowed a maximum of 20 adjusting variables per model.

The only case in which we did not fit every possible model given the available metadata was for one of the T2D cohorts, which, given the number of potential adjusting variables, yielded millions of possible models. Given that we were to vibrate over many thousands of features associated in our initial meta-analysis with T2D, we found computing so many models for each one to be computationally intractable. As such, we selected, for each feature, 10,000 models to fit at random.

## Identification of adjusting variables that heavily confounded associations

For each disease, we modeled the association between the presence or absence of a given adjuster and the change in the absolute value of the average beta coefficient on the independent, binary disease variable across all microbial features. To account for shifts in the average association between a feature and disease, we used a mixed effect model, with a random effect for individual taxa (Eq 1, *n* corresponds to the number of possible adjusters for a given disease, and *i* corresponds to the number of taxa investigated in relation to that disease).

$$abs(estimate\_on\_disease\_variable) \sim adjuster_1 + \ldots + adjuster_n + (1|taxon_{1..i}) \qquad (1)$$

The resulting estimate size on each of the binary adjuster variables in Eq 1 indicates the change in average microbial_feature ~ disease association when a given adjuster is present. These values are reported in the bar plots in Fig 4.

## Benchmarking data transformations, adjusting variables, and vibration numbers

We used the T2D datasets (which have, in total, the most number of potential adjusting variables recorded) to compare the impact of different data transformations, adjusting variables, and vibration numbers on our results. We compared center logged ratio (CLR) transformations on each dataset, our logging strategy described above, and the raw abundance data, running our entire pipeline with 10,0000 vibrations. We additionally compared our results when 3, 6, or 9 variables were selected at random from each cohort.

As a measure of robustness, we computed the fraction of all associations that were positive, with fractions approaching 1 or 0 being highly robust, and fractions approaching 0.5 being nonrobust (e.g., reporting conflicting results close to half of the time). For **S2D Fig**, we computed the Pearson correlation between these fractions.

Finally, we additionally compared the results of the mixed effects confounder analysis as a function of number of vibrations fit. In **S4 Fig**, we report the output of this analysis and the correlation between the estimate impact of each adjusting variable on model output.

## Plotting and figure generation

All plots were made with R's ggplot2 [52] package. We assembled figures in Adobe Illustrator.

## Other software information

All statistical analyses were conducted in R. We ran the VoE pipeline on Harvard Research Computing's O2 system.

## Supporting information

**S1 Table. Cohort summary statistics and adjusters present for each disease and references for species of interest from associations in the literature.** This figure can be generated using the code deposited in https://github.com/chiragjp/ubiome_robustness and the data deposited in https://figshare.com/projects/Microbiome_robustness/127607.
(XLSX)

**S2 Table. Output of initial associations and summary of VoE for findings reported in the literature.** This figure can be generated using the code deposited in https://github.com/chiragjp/ubiome_robustness and the data deposited in https://figshare.com/projects/Microbiome_robustness/127607. VoE, vibration of effects.
(XLSX)

**S1 Fig. Benchmarking data transformations and the number of adjusting variables using our T2D datasets. (A–C)** The impact of the different numbers of vibrations and data transformation methods on VoE. We plot the number of features that were FDR significant at least once upon vibration with different numbers of adjusting variables considered as well as different data transformation strategies (i.e., logged versus raw abundances versus center log ratio transformations). **(D)** The number of $p$-value and **(E)** FDR significant findings for the T2D cohort with the largest number of possible adjusters (using only log-transformed data, as opposed to the previous 3 panels). **(F)** The robustness of associations as a function of number of vibration variables and modeling strategy. We computed the fraction of associations that were positive for any given microbial feature—a highly robust association is 100% positive or 0% positive (i.e., negative), whereas a nonrobust association is closer to 50% positive (i.e., inconsistent in direction). In this heatmap, we correlated these associations for all features to gauge if the different data transformations and numbers of adjusting variables yielded similar measures of robustness across all datasets. This figure can be generated using the code deposited in https://github.com/chiragjp/ubiome_robustness and the data deposited in https://figshare.com/projects/Microbiome_robustness/127607. FDR, false discovery rate; T2D, type 2 diabetes; VoE, vibration of effects.
(PDF)

**S2 Fig. VoE for reported associations from the literature. Bolded species names on the x-axis correspond to organisms that were FDR significant in our study. N corresponds to the**

**number of models that converged for a given organism.** The middle bar describes the fraction of association sizes greater than 0 for a given association: A highly confounded association will be closer to 0.5 and pink, whereas more robust associations will be closer to 0 or 1 and blue. The gray bars in upper bar plot corresponds the fraction of models that were nominally (*p*-value < 0.05) significant for the microbial feature–disease association, whereas the black bars correspond to the fraction of models that were FDR significant. This figure can be generated using the code deposited in https://github.com/chiragjp/ubiome_robustness and the data deposited in https://figshare.com/projects/Microbiome_robustness/127607. FDR, false discovery rate; VoE, vibration of effects.
(PDF)

**S3 Fig. The association between *Roseburia* and ACVD, which we could have overlooked if only fitting a single modeling strategy.** Each point represents a different model specification, the x-axis is the beta coefficient on the binary disease variable, the y-axis is the −log10(*p*-value). The dotted line represents FDR adjusted significance. The solid line represents nominal significance. This figure can be generated using the code deposited in https://github.com/chiragjp/ubiome_robustness and the data deposited in https://figshare.com/projects/Microbiome_robustness/127607. ACVD, atherosclerotic cardiovascular disease; FDR, false discovery rate.
(PDF)

**S4 Fig. Benchmarking the number of vibrations needed to estimate the effect of confounding on microbiome associations. (A)** The output of our confounder analysis (e.g., in Fig 4). The x-axis is the number of vibrations. The y-axis is each possible adjusting variable in the T2D associations. The values correspond to the beta coefficient (from our mixed effects analysis) describing the average change in microbiome–disease associations when a given adjusting variable is present in a model. **(B)** The correlation between the values in panel A at different numbers of vibrations. This figure can be generated using the code deposited in https://github.com/chiragjp/ubiome_robustness and the data deposited in https://figshare.com/projects/Microbiome_robustness/127607. T2D, type 2 diabetes.
(PDF)

**S1 Text. The impact of the number of adjusting variables and data transformation method on VoE. VoE, vibration of effects.**
(DOCX)

# Acknowledgments

We thank Harvard Research Computing for providing compute resources for this work.

# Author Contributions

**Conceptualization:** Braden T. Tierney, Aleksandar D. Kostic, Chirag J. Patel.

**Data curation:** Braden T. Tierney, Yingxuan Tan, Zhen Yang, Bing Shui, Michaela J. Walker, Benjamin M. Kent.

**Formal analysis:** Braden T. Tierney, Yingxuan Tan.

**Funding acquisition:** Chirag J. Patel.

**Investigation:** Braden T. Tierney, Yingxuan Tan, Zhen Yang, Bing Shui, Michaela J. Walker, Benjamin M. Kent, Chirag J. Patel.

**Methodology:** Braden T. Tierney, Chirag J. Patel.

**Project administration:** Braden T. Tierney, Aleksandar D. Kostic, Chirag J. Patel.

**Resources:** Aleksandar D. Kostic, Chirag J. Patel.

**Software:** Braden T. Tierney, Yingxuan Tan.

**Supervision:** Braden T. Tierney, Aleksandar D. Kostic, Chirag J. Patel.

**Validation:** Braden T. Tierney.

**Visualization:** Braden T. Tierney.

**Writing – original draft:** Braden T. Tierney, Aleksandar D. Kostic, Chirag J. Patel.

**Writing – review & editing:** Braden T. Tierney, Aleksandar D. Kostic, Chirag J. Patel.

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
