## [Editor Report · Decision Letter 0]

7 Jul 2021

Dear Chirag, 

Thank you for submitting your manuscript entitled "Evaluating 581 reported gut microbiome-disease associations across 1.1 million model specifications reveals sources of inconsistency in metagenomic research" for consideration as a Meta-Research Article by PLOS Biology.

Your manuscript has now been evaluated by the PLOS Biology editorial staff, as well as by an academic editor with relevant expertise, and I'm writing to let you know that we would like to send your submission out for external peer review.

Please re-submit your manuscript within two working days, i.e. by Jul 09 2021 11:59PM.

Kind regards,

Roli

Roland Roberts

Senior Editor

PLOS Biology

rroberts@plos.org

---

## [Decision Letter · Decision Letter 1]

20 Aug 2021

Dear Dr. Patel,

Thank you for submitting your manuscript "Evaluating 581 reported gut microbiome-disease associations across 1.1 million model specifications reveals sources of inconsistency in metagenomic research" for consideration as a Meta-Research Article at PLOS Biology. Your manuscript has been evaluated by the PLOS Biology editors, an Academic Editor with relevant expertise, and by several independent reviewers.

You will see that reviewer #1 has a large number of queries, regretting your failure to pre-register the study, asking you to tone down some conclusions, and looking for improvements in both text and presentation. Reviewer #2 raises a large number of questions about the method, and some specific points about diseases and features of the microbiome.

In light of the reviews (below), we will not be able to accept the current version of the manuscript, but we would welcome re-submission of a much-revised version that takes into account the reviewers' comments. We cannot make any decision about publication until we have seen the revised manuscript and your response to the reviewers' comments. Your revised manuscript is also likely to be sent for further evaluation by the reviewers.

We expect to receive your revised manuscript within 3 months. 

**IMPORTANT - SUBMITTING YOUR REVISION**

*Re-submission Checklist*

*Published Peer Review*

*PLOS Data Policy*

*Blot and Gel Data Policy*

Sincerely,

Roland Roberts

Senior Editor

PLOS Biology

rroberts@plos.org

REVIEWS:

Reviewer #1: Florian NAUDET

Reviewer #2: Katie Pollard & Annamarie Bustion

Reviewer #1: I think that this manuscript has the potential to become an important contribution to the field. I have however mixed feelings about some aspects that I will try to detail in my review. 

Please note that I am also reviewer for another paper of this team in PLOS Biology and that I was also very enthusiastic about this other paper. Please also note that I am a reviewer involved in meta-research and that I strongly suggest that a reviewer involved in the specific field of microbiome research should be invited (a reviewer used to the methods used in this field). 

First of all, this is a stellar piece of research performing various "case studies", applying the concept of vibration of effects in the field of gut microbiome disease associations. VoE is a form of multiverse analysis and is the subject of many research efforts across the world and the team submitting this paper is at the forefront of these efforts. The microbiome field is rapidly expanding and may have serious problem of reproduciblity due to somewhat small sample size and an universe of researcher's degrees of freedom. The application of VoE concept in this case is therefore timely and innovative. One may also note that VoE is proposed here in a very positive manner, as a tool to identify associations worth of interest and not as a "brute force" tool to disqualify any positive association. This paper represent an heroic effort not solely in running various models but also in being an attempt to synthesize certain aspects the field of microbiome research. It is, in my opinion an important contribution to the field of meta-research. 

I however have two major concerns :

- First the research was not pre-registered (or perhaps I missed it). Of course, authors may acknowledge that this is not an hypothesis generating study. But, in my opinion, the paper 

 presents many specific examples as cases studies and there is room for cheery picking. I don't say that authors cherry picked the results but, rather that, without pre-registration, one cannot affirm that it was not the case ; 

. Several choices were made in choosing the examples/case studies. Pre registration would have added more transparency on these choices ; 

. I regret that the authors didn't adopt a systematic review approach : 

A/ It could have helped again in providing more transparency concerning selection of the included topics / case studies ;

B/ Some aspect of the systematic review approach may have been useful to improve both reporting and reproducibility (e.g. a PRISMA flow chart may have been much more intuitive and easy to follow that the current information reported in the various web appendices) ; 

. I anticipate that authors will say that registration may not be mandatory because this is not an hypothesis testing study, and I can understand this point of view. But I also think that there are some degrees of freedom in their study and that it is always best to fix those a priori, as far as it is possible of course ; 

. At the very least, non-registration must be explicitly stated in the methods / and / possible shortcomings must be discussed in the limitations ; 

. It seems that the authors used a protocol (p.13, l.377) but they do not provide this protocol / I would be interested in a specific pargraph detaillin any change to the protocol;

- My second major concern is rather conceptual : 

. Authors suggest that their approach is an advance in identifying association worth of interest. At least this is the tone of the discussion and the conclusion. I do think that there is some spin here and that the design cannot garantee this. 

. Indeed the present study nicely details a path for identifying associations that can be considered as robust regarding various criteria ; 

. But it does not mean that the robust associations identified will necessarily be transformed in more robust discoveries ; 

. It does not explore nor compare with other methods, how much of these associations will end in being false positive / false negative. 

. At the very least the discussion section should be toned done, and further developments necessary to develop and adopt this new approach must be suggested;

I leave the editor judge whether these concerns rather qualify for a major revision or for a rejection. I may suggest major revision but I think that an in depth revision of the discussion will be needed.

The paper is clearly written, understandable for a large audience despite many technical aspects. I do think that authors did a good job in this regard. Of course it could be improved and I hope that the following minor comments will be helpful. 

- COI disclosure: I would be interested to know more about the 2 companies (FitBiomics) and Micro Bioscience, and especially how links with these companies may represent a conflicting interest. 

- Authors must give a close attention to their figures : 

. In one hand figures are very nice to read and nicely summarize many very important information ; 

. In the other hand, figures are very difficult to read (for each figure, it took me some times to understand the key concepts as these figures are not so intuitive) ; 

. For example, in Figure 1, information about the search strategy is somewhat hard to understand, especially if you did not have looked at the method section. Figure 1 could be more aligned with the data reported in the text to make its reading easier. 

A/ I appreciate that y axis are in a log scale / however, the figure should provide p-values (on a log scale) rather than log p-values. It would be easier to read. This could apply to all figures. 

B/ In Figure 1 A, there is no information of the y and y axis for the VoE, taxon graphs. I think that it could be difficult to read. 

C/ In figure 1B and 1G, there is no label for the x-axis. The number 1 to 6 could be difficult to follow. I would suggest to think in more user friendly representation.

. For example, in Figure 2 : 

A/ The Figure use the term Janus Effect in its legend. It is of course defined in the figure but it does not appear in the text. I would suggest that authors could add this in the text;

B/ There is no information about the axes (no label for the y nor for the x axes). It could be difficult to follow. I don't have any good suggestion for improving the readility and adding more information about the associations being studied / but I suggest that authors may think in a more user friendly figure ;

. For example, in figure 3A, I would use the same scale for all effect size across all diseases (e.g. using a log scale) ; 

- P.4 l.98 37.8 findings were from the datasets used in this analysis. I would be interested to know more about reproducibility of a finding in the dataset where this finding is from versus in other datasets (e.g. in line 109-110, I would be interested to break this by original versus new datasets). It could be out of scope but I would like to hear the authors'point of view about this point.

- P.4. l.111, I'm not sure that the title is adequate. Instead of "additional potential", I would suggest "different". In this paragraph I would also be interested in Venn's diagram showing the overlap between usual approach and the approach proposed by the authors ;

- p5. l.119 : "we hypothesized" OK : was a it a priori ? Or was it a posteriori (see my previous comment) in major concerns ; 

- p.6. l.159-161. This is rather an interpretation and should be moved in the discussion section. And, also, following my second major concern, authors must provide convincing evidence to support such a statement. I'm not sure that it is possible based on their data. 

- p7. l193: Here again I'm surprised that the author selected a specific example (i.e. F. Prausnitizii). Was this done a priori. Why this specific association? I would have preferred using specific criteria and a random selection of example using the predefined criteria. I may have missed something however. 

. P.8 l.206: I think that the very first sentence of the discussion must summarize the findings. I would therefore suggest to delete (or move in the introduction) the very first sentences. 

. P. 10 l.261-263: In line with my previous comment, I would be interested to know more about discrepancies between re-analysis in a given dataset and the original analysis. 

. p.14, l.422: How the 20 adjusting variables were selected. I think that this point could benefit from more details. 

Reviewer #2: This paper explores the microbiome field's need to account for the portion of bias introduced by confounding variables when choosing models of association between bacterial taxa and human disease. The authors chose to address this through a process called "Vibration of Effects." In addition to identifying associations that change when adjusting for potential confounding variables, the authors promise to:

Measure reproducibility of previous literature findings of association, 

Identify new biological associations between bacterial taxa and human disease,

Provide recommendations on feature prioritization in future studies, based on their ability to recapitulate associations in this work.

Identify whether an adjuster is of the confounder or collider variety.

Major comments: 

The authors state that VoE can be used to assess reproducibility, but the most non-robust associations were also those that had the most available covariates, and therefore yielded the highest number of models. This process seems highly dependent on the type and number of adjusters available in sequencing metadata. VoE seems better suited to evaluating associations after a study without attempting to narrow in on a one, best model. 

In addition to the number of measured covariates, VoE appears to be sensitive to the number of cohorts, sample size, and variability in measurements. These differ across the studied diseases. When comparing results over diseases (Lines 140-145, for example) these factors should be discussed even more explicitly. Single-cohort associations are mentioned, but it is not clear if cohort is one of the covariates included in VoE or could be included. Providing simulations or down-sampling experiments on the larger data sets to show how these variables affect VoE in this application would be extremely useful. 

The premise for using VoE needs further justification. Why would a marginal association be inherently better if it differs little from all possible conditional associations? Identifying confounding variables makes sense, but it seems like this method is broader than that. Justification needs to be provided for seeking reproducibility over models with all possible/measured variables given that the authors acknowledge that some covariates should not be adjusted for. Similarly, why is a microbe-disease association inherently more interesting if it is reproduced across multiple diseases (Lines 226-228 for example)?

An association-prioritization framework of variable adjusters that must be considered across multiple diseases and studies is promised as a result, but not provided by the manuscript. 

How are the findings and measurements of statistical significance affected by only analyzing previously significant microbe-disease associations (Lines 252-257)? Regression to the mean type thinking and extreme value statistics suggest that these will be biased towards non-significance and smaller effect sizes. It would be interesting to include non-significant results and look at how many become more significant. What is known about VoE conditional on starting with previously reported associations versus vomiting this conditioning? 

Other comments:

Line 63: Explain VoE in more detail (text on Lines 419-422 seems helpful, for example) and any statistical theory justifying it as a metric for evaluating measured associations. Is there a causal inference framework underlying VoE?

Line 85/102-110: Using a baseline with no covariates seems simplistic compared to the modeling used in the published studies. It would be great if the authors included whether or not their baseline model matched the models explored in the original studies. That is, it doesn't seem fair to mount the comparison in Figure 1B, unless the reader is assured that the baseline model is a fair point of reference.

Line 93: Provide a citation for the existing database, or use a different word here.

Line 96: Rheumatoid arthritis and obesity might be additional indications to explore. Related, IBD could be stratified by UC and CD.

Line 112: Provide more detail about the models in terms of parametric forms, outcomes, and covariates. 

Lines 119-130: Figures 1B and 1G do not seem dissimilar, yet the authors state that VoE helped recover reported results lost by the baseline model. The purpose of the baseline model as a comparison merits discussion here. In Figure 1G, might it be better to report the proportion of findings matched, rather than the median association from all models generated?

Line 131: Could association robustness also be varying as a function of number of adjusters, rather than disease/cohort? T1D and T2D are reported to be the least robust, but these studies also resulted in the most models generated due to their high number of available covariates.

Line 148: What is an arbitrary model? It is not clear if this statement refers to a null distribution or probability under the empirical distribution or something else? 

Line 150/296: Figure 2 does not have an A and B panel. Also, is the baseline comparison necessary to include?

Line 157-159: It would be helpful to explain why the marginal association is not significant. 

Lines 155-161/337/403: The previously reported association between Roseburia and ACVD was found via an ACVD-microbiome association based on gene and KEGG content, rather than taxa. We wonder if the writers of this manuscript have considered exploring more than relative abundance of species. Gene content and pathway presence/absence could also be explored. Related, because this is already a known association, we are assuming the authors believe this to be potentially overlooked based on their baseline model in this work. 

Lines 162-192: The authors present the influence of adjuster variables on these particular datasets, but they do not show that these can be prioritized generally in a so-called association-prioritization framework across multiple studies and diseases.

Lines 201-203: Is this a reference to confounders, colliders, or something else? 

Lines 226-228: We disagree that an association-prioritization framework has been provided, but maybe VoE could be applied in the future to do so. In this work, there do not seem to be adjusters that are consistently significant across multiple diseases and studies.

Line 245: Geography didn't seem significant for F. prausnitzii associations.

Line 256/405: It might make sense to pick one time point from longitudinal studies, rather than averaging.

Line 258/403: We recommend performing CLR before comparing abundances across multiple studies of the same disease.

Line 311: We believe the authors meant to say "y-axis."

Line 312: Figure 3 does not have colors for adjusters used across multiple diseases, as written.

Line 325: Needs a y-axis label

Line 371: What is the justification for using the second publication? 

Line 402: We expected the models to be set up as disease~microbial feature + covariates. What is the justification for making the microbial feature the outcome? 

Line 428: 10,000 is a lot of models. This data set could be used to explore how the number of covariates affects the number of robust associations. What happens if a random 10, 100, 1000 models are used? Or if all models for a random subset of covariates are used? 

Style comments:

In general, there is overuse of quotation marks throughout the manuscript that do not enhance understanding. Examples: "consistent" on line 25, "non-robust" on line 83, "recovered" on line 84, etc.

Line 59-62: This is a run-on sentence that hinders understanding.

Line 86: The phrase after the colon is missing words.

It is our policy to sign reviews: Annamarie Bustion and Katie Pollard

---

## [Decision Letter · Decision Letter 2]

17 Jan 2022

Dear Chirag,

Thank you for submitting your revised Meta-Research Article entitled "Evaluating 581 reported gut microbiome-disease associations across millions of model specifications reveals sources of inconsistency in metagenomic research" for publication in PLOS Biology. I have now obtained advice from the original reviewers and have discussed their comments with the Academic Editor. 

Based on the reviews, we will probably accept this manuscript for publication, provided you satisfactorily address the remaining points raised by the reviewers. Please also make sure to address the following data and other policy-related requests.

IMPORTANT:

a) Please attend to the remaining requests from reviewer(s) #2.

b) Please re-name your supplementary Fig files "S1_Fig," "S2_Fig," etc.

c) Please address my Data Policy requests below; specifically, we need you to supply the numerical values underlying Figs 1BCDEF, 2AB, 3, 4ABCDEFGH, 5ABC, S1ABCD, S2, S3, S4AB. If these can all be generated from the data and code deposited in Github/Figshare, please cite the location of the data clearly in each relevant main and supplementary Fig legend, e.g. “This Figure can be generated using the data and code deposited in https://github.com/chiragjp/ubiome_reproducibility”).

We expect to receive your revised manuscript within two weeks. 

*Published Peer Review History*

*Early Version*

Sincerely,

Roli

Senior Editor,

rroberts@plos.org,

PLOS Biology

DATA POLICY:

Regardless of the method selected, please ensure that you provide the individual numerical values that underlie the summary data displayed in the following figure panels as they are essential for readers to assess your analysis and to reproduce it: Figs 1BCDEF, 2AB, 3, 4ABCDEFGH, 5ABC, S1ABCD, S2, S3, S4AB. NOTE: the numerical data provided should include all replicates AND the way in which the plotted mean and errors were derived (it should not present only the mean/average values).

DATA NOT SHOWN?

REVIEWERS' COMMENTS:

Reviewer #1:

[identifies himself as Florian Naudet]

Thank you for your response. I found these responses very appropriate and all my previous concerns are now well addressed. 

I really think that this manuscript is an important contribution to the field of meta-research. 

Reviewer #2:

[identify themselves as Annamarie Bustion and Katie Pollard]

The paper is improved; the results are presented more clearly and the conclusions drawn are more metered. The authors re-identified their research goals as the following: identification of confounders, recovery and robustness assessments of previous literature findings, and identification of new associations. The text answers these goals, and the authors appropriately de-emphasized their method's ability to assess reproducibility or provide an umbrella feature prioritization framework. Also, the authors present the rationale for using VoE more comprehensively, and they better describe their rationale for using a simple baseline model rather than recapitulating the models from their literature findings.

Remaining critiques:

Authors' response to 2.1: The authors now provide an assessment of VoE sensitivity to number of variables available, using T2D as an example. This does provide confidence that VoE can provide information on non-robust associations independent of number of variables. However, a stronger response would have looked at larger values for the number of variables, not just a comparison of 3, 6, and 9 variables. Further, an assessment of only three variable sizes seems insufficient to state that there could be a "a potential leveling-off with increasing adjusting variables." We recommend adding results for some higher numbers of variables. 

Authors' response to 2.2: It is now clear how the authors made use of multiple cohorts. But it would have been useful for the authors to instead use additional disease cohorts as a means of external validation in addition to the use of previous literature findings. If possible, we recommend adding this analysis. 

Abstract: Before reading the main text, it is not clear what "model specifications" refers to. This could be different covariates, a different parametric form, or a data transformation, amongst other things. It would be very helpful to clearly state that the focus of this paper is exploring all the possible combinations of covariates. Note: In the Introduction (L74-76), this is pretty clear. Our recommendation is to include a similar definition / scope statement in the abstract. 

Abstract/Introduction: We appreciate that the authors now also look at non-associations with VoE. We recommend that you check the places where you talk about focusing on 581 published associations: some of the text sounds like you only analyze the associations but not the non-associations. Alternative language might be "studies" or "cohorts" "in which disease-microbiome associations were previously detected". 

Figure 2A: The provided Upset plot is not fully explained. What do the gray bars represent? 

Figure 2B: These results are more interpretable now that the authors are exploring fraction of matching associations. But these graphs would be more legible as count histograms rather than density plots.

It is our policy to sign reviews: Annamarie Bustion and Katie Pollard

---

## [Editor Report · Decision Letter 3]

27 Jan 2022

Dear Chirag,

On behalf of my colleagues and the two Academic Editors, Jonathan Eisen and Marcus Munafo, I'm pleased to say that we can in principle accept your Meta-Research Article "Systematically assessing microbiome-disease associations identifies drivers of inconsistency in metagenomic research" for publication in PLOS Biology, provided you address any remaining formatting and reporting issues. These will be detailed in an email that will follow this letter and that you will usually receive within 2-3 business days, during which time no action is required from you. Please note that we will not be able to formally accept your manuscript and schedule it for publication until you have any requested changes.

PRESS: We frequently collaborate with press offices. If your institution or institutions have a press office, please notify them about your upcoming paper at this point, to enable them to help maximise its impact. If the press office is planning to promote your findings, we would be grateful if they could coordinate with biologypress@plos.org. If you have not yet opted out of the early version process, we ask that you notify us immediately of any press plans so that we may do so on your behalf.

Sincerely, 

Roli

Roland G Roberts, PhD 

Senior Editor 

PLOS Biology

rroberts@plos.org